# Insecticides and Drought as a Fatal Combination for a Stream Macroinvertebrate Assemblage in a Catchment Area Exploited by Large-Scale Agriculture

**Marek Let [1],\*, Jan Špaček [2], Martin Ferenčík [2,3] , Antonín Kouba [1] and Martin Bláha [1],\***

[1] South Bohemian Research Center of Aquaculture and Biodiversity of Hydrocenoses, Faculty of Fisheries and Protection of Waters, University of South Bohemia in České Budějovice, Zátiší 728/II, 389 25 Vodňany, Czech Republic; akouba@frov.jcu.cz

[2] Povodí Labe, Víta Nejedlého 951/8, Slezské Předměstí, 500 03 Hradec Králové, Czech Republic; spacekj@pla.cz (J.Š.); ferencikm@pla.cz (M.F.)

[3] Institute of Environmental and Chemical Engineering, Faculty of Chemical Technology, University of Pardubice, Studentská 573, 532 10 Pardubice, Czech Republic

\* Correspondence: mlet@frov.jcu.cz (M.L.); blaha@frov.jcu.cz (M.B.)

**Abstract:** This case study documents responses in a headwater macroinvertebrate assemblage to insecticide pollution and hydrological drought. In 2014, the Doubravka brook (Czech Republic) was damaged by a large overflow of a mixture of chlorpyrifos (CPS) and cypermethrin (CP). In 2016–2017, this brook was then affected by severe drought that sometimes led to an almost complete absence of surface water. We found significant relationships between the strength of both these disturbances and the deeper taxonomic levels of both the overall macroinvertebrate assemblage (classes) and the arthropod assemblage alone (orders and dipteran families), as well as the functional feeding groups (FFGs). The CPS-CP contamination was mostly negatively correlated to arthropod and non-arthropod taxa and was positively correlated only with FFG collector-gatherers; on the other hand, the drought was negatively correlated to Simuliidae, Ephemeroptera, Trichoptera, and the FFG of grazer-scrapers and passive filterers. Drought conditions correlated most positively with Isopoda, Ostracoda, Heteroptera, adult Coleoptera, and predator and active filterer FFGs. The chosen eco-indicators (SPEAR$_{pesticides}$, SPEAR$_{refuge}$, BMWP, and EPT) used as support information reveal the poor ecological status of the whole assemblage, including the control site, the cause of which is most likely to be the exploitation of the adjacent catchment area by large-scale agriculture. This type of agricultural exploitation will undoubtedly affect macroinvertebrate assemblages as a result of agrochemical and soil inputs during run-off events and will also exacerbate the effect of droughts when precipitation levels drop.

**Keywords:** headwaters; benthic species; chlorpyrifos; cypermethrin; organophosphate insecticide; pyrethroid insecticide; hydrological droughts; functional feeding groups; contamination

## 1. Introduction

Macroinvertebrate assemblages inhabiting headwaters in a cultural landscape have to cope with exposure to anthropogenic activities and their implications [1–5]. The vast majority of published studies generally associate human activities with negative impacts on stream invertebrate biota and their ecological functioning; that is, they provoke a fall in diversity, abundance, biomass, and organic matter processing [6–12]. Nevertheless, human-triggered effects can result in greater biomass of specific aquatic organisms [13–15] or, occasionally, in higher species richness [16,17], and some ostensibly heavily affected sites may sometimes even act as refuges for endangered species [18–20].

Over the last few years, much of Europe has suffered from severe droughts in combination with unusually high temperatures [21,22], and climatological prognoses suggest that these events will become more frequent in the future [23,24]. Combined with intensive

land use and its effects on functioning water regimes, climate change could cause huge ecological and economic damage [25–28]. Affected headwaters will dry up more often and more quickly, thereby multiplying the impact of pollution by agricultural or urban wastewater [29–31]. Aside from the plethora of pollutants that occur in surface waters, pesticides (e.g., insecticides, herbicides, and fungicides) are of concern due to the negative effect they have, in particular, on aquatic biota and, in general, on whole ecosystem functioning [32–35]. The main reason is their massive use worldwide and their typically high specific toxicity for non-target aquatic organisms [5,36–38]. Pesticides are an integral part of conventional agriculture and enter the water at much higher concentrations after short-term runoff events than those that are usually detected by standard monitoring sampling methods [39,40]. Therefore, pesticides or their residues may have acute or chronic effects on aquatic organisms [41,42]. Furthermore, toxicants with great adsorption to soil or organic matter may persist in sediments; hence, they can threat sediment dwelling organism for a prolonged time [10,43]. The presence of pesticides in aquatic ecosystems is often accompanied by a wide range of anthropogenic influences, such as habitat degradation, drainage [44,45], artificial siltation and sedimentation [46,47], increased nutrient input, and the occurrence of more frequent extreme hydrological events (droughts and floods) at interconnected sites [44,48]. These factors often exacerbate the negative effects of pollutants on aquatic biota, generally due to the losses they induce in many kinds of resources in an ecological sense (e.g., microhabitats, refuges, food, and the strength of interactions) [49–51].

Invertebrate populations should be able to withstand specific disturbances whose impact will presumptively be reversible. Nevertheless, certain regularly recurring processes in combination with other biotic or abiotic conditions will undoubtedly damage exposed assemblages by eliminating the most susceptible species (e.g., *Margaritifera margaritifera* or *Prosopistoma pennigerum*) or even entire deeper taxonomic groups (e.g., heptageniid mayflies, perlid stoneflies, and goerid caddisflies), thereby eventually negating ecological traits [4,49,52–57]. For this reason, laboratory and field experiments such as those defined by Diamond [58] that manipulate the effects of xenobiotics under various experimental settings (simulation of diverse possible natural scenarios) are necessary [11]. However, experiments carried out in natural scenarios (natural experiments) are also essential as they can verify obtained knowledge in real environments [59].

The aim of this study was thus to assess the effects of serious insecticide contamination on a headwater macroinvertebrate assemblage and observe the ability of species to recolonize affected stretches of the damaged water course. We used a dataset that was not created purely for research purposes; rather, it was used by the state environmental protection authorities as part of their legal investigation into the accident. Thus, it lacks some of the information needed for a thorough assessment of certain aspects of the macroinvertebrate assemblage. Our study reveals the response of this macroinvertebrate assemblage to an extreme event, a situation that is rarely observed, probably because—unlike chronic effects—these accidents only occur rarely. In terms of climate change and evolving strategies in agriculture, our results reflect a situation that will increasingly threaten organisms in watercourses in agriculture landscapes, as well as all other lifeforms that are dependent on these damaged water bodies.

## 2. Materials and Methods

### 2.1. Locality Description

The Doubravka brook is situated in central Bohemia (Czech Republic, Central Europe, spring to confluence: N 49.7775686, E 15.5793436–N 49.8619361, E 15.4977614; Figure 1). This small, third-order stream (the Stahler order used) is 13.8 km long, has a catchment area of 21.6 km$^2$ and average discharge of up to 9 L s$^{-1}$. It rises at an altitude of 460 m a.s.l. and flows into a fifth-order stream at 251 m a.s.l. The riverbed of the brook is natural and it is surrounded by typical alluvial vegetation, together with part of a cultural coniferous forest (in all, an approximately 150–600 m wide bio-corridor). This sector of the brook has been preserved due to the steep terrain. Most of its catchment area is occupied by

uniform blocks of fields (each with an average area of 30 ha) where rapeseed, corn, and cereals are cultivated. The fields are drained by underground water collectors connected to straight channels, as well as to artificial and/or originally temporary or permanent tributaries. The brook ecosystem harbors microhabitats with shallow riffles with macro-, meso-, micro-lithal, and pools anchored by the root systems of alder trees (*Alnus glutinosa*, *A. incana*). In these microhabitats detritus, particulate organic matter (POM), smaller-sized pieces of gravel and xylal accumulate. However, in recent years this locality has been affected by severe droughts leading to zero surface discharge, a phenomenon that was often observed during the study period, mainly during extremely hot summers. The riffles often disappeared and the bottom of the isolated pools became almost completely covered by silt and organic matter.

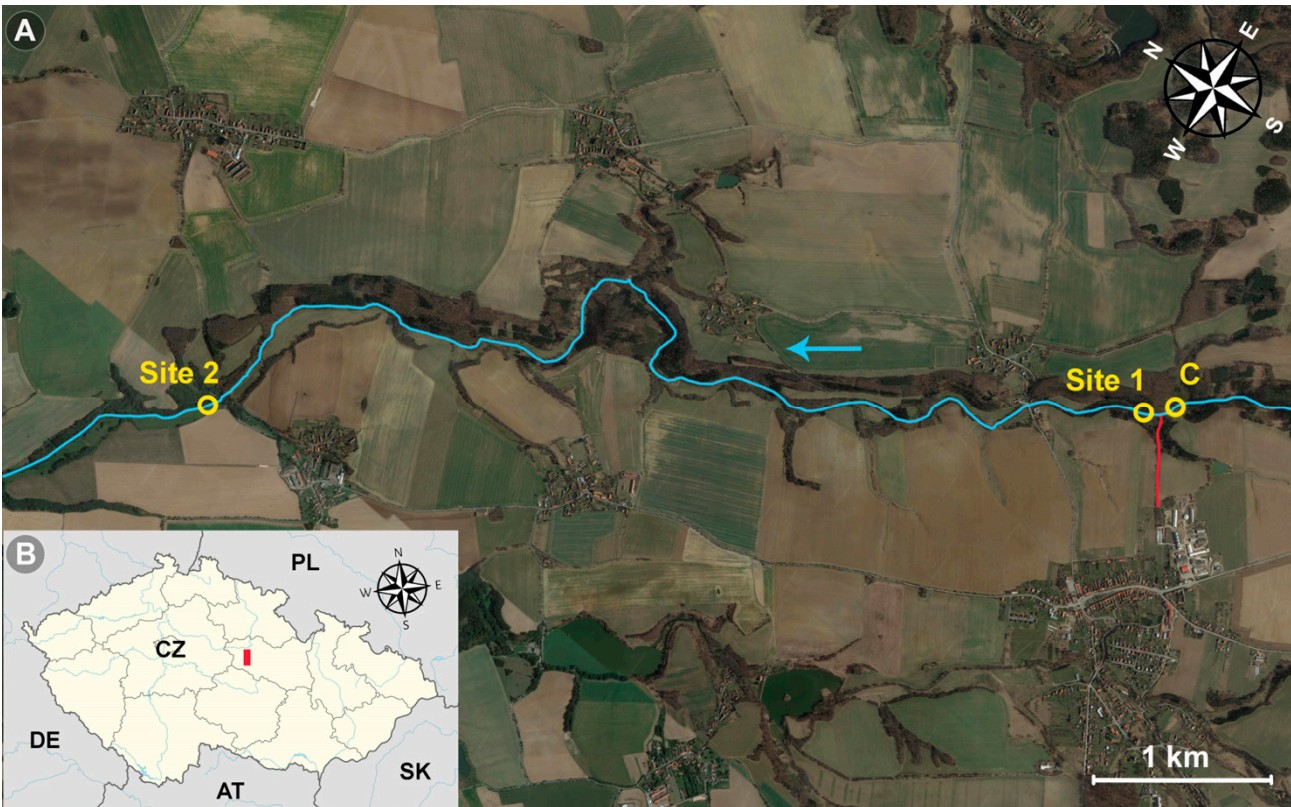

**Figure 1.** Map of the study area: (**A**) Locations of the sampling sites along the longitudinal profile of the Doubravka brook (in blue; flow direction indicated by the blue arrow). The contaminated drainage channel is highlighted in red. The sampling sites are shown as yellow circles: C and Site 1 = Transect 1; Site 2 = Transect 2. (**B**) Study area shown as the red rectangle in the Czech Republic.

### 2.2. Insecticide Contamination

In March 2014, the brook was accidentally contaminated by a commercial insecticide product containing organophosphate chlorpyrifos (CPS) and pyrethroid cypermethrin (CP) (at a ratio of 10:1, respectively). It flowed down a straight drainage channel receiving wastewater and runoff-water from a local agricultural company that had illegally stored unused insecticide. The concentration of CPS in the sediment dry weight (d.w.) was 13 mg kg$^{-1}$ at the confluence of the brook and the drainage channel (Site 1, Figure 1) five days after the accident. Mass mortality of over 10,000 individuals of the critically endangered noble crayfish *Astacus astacus*, macrozoobenthos, and fish (brown trout *Salmo trutta* and stone loach *Barbatula barbatula*) was observed for over 6 km downstream from Site 1. *A. astacus* and *B. barbatula* samples were analysed for CPS (77 and 31,000 µg of CPS kg$^{-1}$ of body weight, respectively). CPS sediment concentrations decreased rapidly

over the following months (Table 1). The bottom of the contaminated stretch of the brook also became abnormally overgrown by filamentous algae and periphyton during the first months after the accident.

**Table 1.** Chlorpyrifos content in the sediments analysed in the samples from Sites C (control = unaffected, Transect 1), 1 (affected, Transect 1) and 2 (affected, Transect 2) during the sampling period 2014–2017. The asterisks indicate macroinvertebrate sampling. LOQ = 20 µg kg$^{-1}$ (d.w.).

| Sampling Date | Site C (µg kg$^{-1}$ d.w.) | Site 1 (µg kg$^{-1}$ d.w.) | Site 2 (µg kg$^{-1}$ d.w.) | Time Passed after Accident (Days) |
|---|---|---|---|---|
| 4th IV 2014 | <LOQ | 13,000 | 350 | 5 |
| 20th VI 2014 * | <LOQ | 128 | 51 | 83 |
| 17th IV 2015 * | <LOQ | 87 | <LOQ | 384 |
| 26th VII 2016 * | <LOQ | 33 | <LOQ | 850 |
| 19th IX 2017 * | <LOQ | 25 | <LOQ | 1476 |

*2.3. Sediment Sampling*

Sediment was sampled according to the standardized accredited methods (ČSN EN ISO 5667-1, ČSN EN ISO 5667-3, ČSN ISO 5667-12, ČSN ISO 5667-14, and ČSN ISO 5667-15) by a telescopic sampler with a wide-neck stainless steel container. A layer of the bottom sediment was scrapped using the edge of the container. Regarding the character of the riverbed containing a small amount of fine sediment, sediment was sampled in several places where it was possible within one sampling site. These individual sub-samples were homogenized into one mixed sample, put inside polyethylene sampling bottles, and stored in a polystyrene thermo box filled with ice during transport prior to the laboratory analysis.

*2.4. Analysis for Pesticides*

2.4.1. Reagents and Materials

The analytical standards of chlorpyrifos (chlorpyrifos-ethyl, CAS 2921-88-2) and deuterated chlorpyrifos-D10 (CAS 285138-81-0) (Dr. Ehrenstorfer; Augsburg, Germany) were of 99.49% and 99.1% purity, respectively. Methanol and mobile phase additive ammonium acetate were LC-MS purity (Merck; Darmstadt, Germany). An SG Water Ultra Clear TWF UV plus TM with TOC-Monitoring water system from SG Water (Hamburg, Germany) was used throughout the study to obtain the LC-MS grade water.

2.4.2. Sample Processing and Analyses

The ultrasonic assisted extraction with methanol solvent was applied for the preparation of sediment samples as it was described by Ferenčík and Schovánková [60]. The samples were dried at 20 °C using a lyophilizer Christ Alpha 1–4 (Osterode, Germany), and sifted through a 2 mm Retsch sieve (stainless steel). The 10 mL aliquot of each sample was homogenized by grinding using a mixer mill MM 200 from Retsch (Haan, Germany) and zirconium oxide 25 mL mixing jars with balls. 5 µL of the internal standard solution (chlorpyrifos-D10, 100 pg mL$^{-1}$) were spiked into 0.5 g homogenized subsamples, then extracted using 5 mL of methanol in 50 mL glass Erlenmeyer flasks in ultrasonic bath Binder Electronic (Berlin, Germany) for 60 min at 40 °C (maximum ultrasonic power). The extracts were centrifuged by centrifuge Jouan B4I (Thermo Electron Industries, Chateau-Gontier, France) at 2400 g. The 100 µL aliquot of extract was diluted with 900 µL of 100 mmol L$^{-1}$ ammonium acetate in a 2 mL sample vial used for LC-MS/MS measurement. A smaller aliquot of a sample or higher extraction volume was selected for the more contaminated samples.

2.4.3. LC-MS/MS Conditions

Based on published literature [61], development and optimization of the LC-MS/MS method was done. The chromatographic separations were conducted using a Waters Acquity UPLC HSS T3 column (100.0 × 2.1 mm, 1.8 µm particle size) and guard col-

umn 2.1 × 5.0 mm with the same chemistry, thermostated at 40 °C. 5% methanol and 5 mmol $L^{-1}$ ammonium acetate in water (A) and 5 mmol $L^{-1}$ ammonium acetate in methanol (B) were used as mobile phases. Gradient elution program was set up from 0.1% B to 99.9% B, the method duration was 18.5 min. The injection volume was 250 μL.

The LC-MS/MS analysis was carried out using triple quadrupole Waters Premier XE (Manchester, UK) connected with ultra-high-performance liquid chromatograph Waters Acquity UPLC® (Milford, MA, USA). The electrospray ionization source heated at 120 °C in the positive ionization mode (3.5 kV) was used. For acquisition and processing of data, the Waters MassLynx software was used.

Chlorpyrifos SRM transitions (precursor > product) were as follows: quantifying 349.8 > 197.7 (cone voltage—25 V and collision energy—19 V), qualifying 349.8 > 96.7 (cone voltage—25 V and collision energy—34 V). In addition, internal standard chlorpyrifos-D10 transitions were as follows: quantifying 359.8 > 198.7 (cone voltage—25 V and collision energy—19 V).

### 2.4.4. Validation of the Analytical Procedure

Six points' calibration curve was linear in the range of 10–500 ng $mL^{-1}$. Internal standard and matrix matching standard methods were used for quantification of chlorpyrifos. Recoveries of chlorpyrifos in sediments were calculated by spiking the matrix at two concentration levels—50 and 500 ng $g^{-1}$—in triplicates. The range of recoveries of the spiked samples was 78–121%. The repeatability of the method was determined as relative standard deviation (RSD) of repeating analysis of spiked samples and was lower than 20%. The method detection and quantitation limits of chlorpyrifos was expressed based on S/N ratio of 3 and 10, respectively [61]. The limit of quantitation (LOQ) was calculated to 20 μg $kg^{-1}$ (d.w.) with a measurement uncertainty of 30% covering the whole sample preparation and measurement procedure assessed using control chart (2 s) constructed at a concentration level 50 μg $kg^{-1}$.

### 2.5. Macrozoobenthos Sampling and Analysis

### 2.5.1. Sampling and Determination

Monitoring took place in 2014–2017. Three sites along the brook were chosen (Figure 1). Site C (control) was not contaminated by the insecticides. Sampling of macrozoobenthos was performed using kick sampler (25 × 25 cm net frame dimensions, 500 μm mesh) by 3 min-long sampling multihabitat method, following rules of European Water Framework Directive (WFD) [62]. Briefly, this method is based on an a priori evaluation of presence and spatial proportions of various microhabitats at the chosen sampling site. Subsequently, an interval taken from the total time (3 min) is allotted for each sampling at particular chosen micro-habitats regarding to their spatial proportionality compared with the other ones. The sub-samples of micro-habitats were always pooled into only one sample, processed using a round steel sieve (40 cm diameter, 500 μm mesh) and preserved in 70% technical ethanol. Organisms were determined to the highest possible taxonomic level; their frequencies in the samples were recorded (Table S6). Before determination, single portions of material were individually separated from smaller particles using a small plastic sieve (500 μm mesh) and tap water.

The first sampling was performed in June 2014 (approximately 2.5 months after the accident). The others in 2015–2017 were restricted between April and September (Table 1). The sampling was initiated by the state monitoring authorities investigating the accident. Therefore, the experimental design lacked replicates of macrozoobenthos samples at individual Sites and timepoints and relied instead on a multihabitat sampling approach.

### 2.5.2. Estimation of Taxa Traits

A dataset containing the ecological and plasticity traits of all detected taxa was created (Table S7). The categories of these ecological traits are represented by the following five subsets: *Functional feeding groups* (Collector-gatherers, Shredders, Xylophages, Active filter-

ers, Passive filters, Predators, and Grazer-scrapers,) *Macro-habitat preference* (Limnophiles, Rheophiles, and Torrenticoles), *Meso-habitat preference* (Epibenthos, Endobenthos, and Hyponeuston), *Micro-habitat preference* (Xylal—tree trunks, branches, and roots; Akal—fine to medium-sized gravel; POM—deposits of fine particulate organic matter; Pelal—mud and sludge; Detritus—deposits of coarse particulate organic matter;, Psammal—sand; Lithal—coarse gravel, cobbles and blocks; Phytal—algae, mosses, and plants) and *Reproductive ability* (<0.5 generation per year, 0.5 generation per year, 1 generation per year, 2 generations per year, and >2 generations per year). The plasticity trait used was *Body length* of adult or final larval instar (<4 mm, 4–7 mm, 8–12 mm, 13–18 mm, 19–25 mm, and >26 mm). The classification was empirically encoded by the proportion of a particular type of trait for detected taxa within each subset category.

### 2.5.3. Calculating Eco-Indicator Parameters

The base parameters (total richness, total abundance, Shannon–Wiener index) commonly used to describe a macroinvertebrate assemblage status were generated in R studio software using the *BiodiversityR* package [63] for all samples. Three types of 'Species At Risk' (SPEAR) classifications (coded by dummy variables) were used to reveal the proportions and sums of the abundances of the most susceptible species, as well as and their richness, as follows: (i) 'SPEAR$_{pesticides}$ (old)' were classified taking into account certain lethal concentration levels and species-specific ecological traits that could be critical for the organism exposed to the pesticides [40]; (ii) the 'SPEAR$_{refuge}$' concept represents the adjustment of the original SPEAR$_{pesticides}$ list with a category of sensitive species with a strong potential for successfully recolonising contaminated stretches of streams and brooks [49]; and (iii) 'SPEAR$_{pesticides}$' are a subset of the 'SPEAR$_{pesticides}$ (old)' that do not belong to the 'SPEAR$_{refuge}$'. SPEAR indices were calculated according to the following adjusted equation:

$$\text{SPEAR index} = \frac{\sum_{i=1}^{n} \log(4xi + 1) \times y}{\sum_{i=1}^{n} \log(4xi + 1)} \tag{1}$$

where *n* is the total number of taxa, *xi* is the abundance of taxon *i*, and *y* is a binary code representing taxon *i* thus: 1 if it belongs to the SPEAR, otherwise 0. The log (4x + 1) transformation of the abundances aims to decrease the influence of populations with mass development [49].

EPT (Ephemeroptera–Plecoptera–Trichoptera) richness, abundance, and relative abundance (%) were calculated because many species groups belonging to these three orders are often very sensitive to several types of disturbances including insecticide pollution. EPT-derived parameters are frequently used as a measure of the quality of the environment and/or the seriousness of a disturbance effect by comparing cases. The EPT abundance usually correlates with the SPEAR abundance.

The original *biological monitoring working party* (BMWP) score and its *average score per taxon* (ASPT), which are based on family presence rather than on abundance, was also derived [64]. The aim of the BMWP score is to indicate the status of organic pollution. Families sensitive to organics are usually more susceptible to insecticides.

### 2.6. Classification of Hydrological Status

The hydrological status semi-quantitative scoring system for implementation in the ordination analysis was designed following Boulton and Lake [53] (Table 2).The prior hydrological status at each sampling point was based on our observations (Figures S3–S5), precipitation and temperature data from previous months for the particular area, and discharge data in the river Doubrava—a second-order river receiving water of the Doubravka brook (data taken from the website of the Czech hydrometeorological institute, Prague, Czech Republic, Table S4 and Figure S2). In 2015–2017, abnormally high temperatures in this area were accompanied by a lack of precipitation. The scores assigned to each sample are shown in Table S5.

**Table 2.** Hydrological status score.

| Description: | Score: |
| --- | --- |
| Fast-flowing | 1 |
| Loss of fast-flowing habitats | 2 |
| Loss of lateral connectivity to stream-edge habitats | 3 |
| Loss of longitudinal connectivity, flow stops, isolated pools form | 4 |
| Pools shrink water quality deteriorates | 5 |
| Total absence of water | 6 |

*2.7. Statistical Analysis*

The dataset containing the abundance of individual taxa as response variables and two environmental predictor variables Chlorpyrifos-Cypermethrin contamination status (CPS-CP) and drought status (Drought), and one covariate (Transect), were analysed using the multivariate Canonical correspondence analysis (CCA) in CANOCO 5 software [65]. Similarly, the chosen deeper taxonomic levels for all detected taxa and arthropods (representing the most diverse and, simultaneously, the most sensitive groups to insecticides) were used as a response in the redundancy analysis (RDA)—classes and orders (dipterans divided into families), respectively. The variables CPS-CP and taxa abundance were log-transformed; the response variables were centered but not standardized. Additionally, the ecological and biological traits of the invertebrates (see Section 2.4.2. Estimation of species traits) were implemented as standardized composition-weighted trait averages (using the abundance of taxa in the samples) into independent RDA against the CPS-CP and Drought variables. The significance of the relationships between these environmental variables and individual compositions of taxa or traits was tested using particular canonical axes and their eigenvalues with a Monte Carlo permutation test (number of permutations: 1999). The cases were always shifted into two permutation blocks, defined by the *Transect* covariate, to maintain autocorrelation. Principal component analyses (PCA) were used for the trait compositions, which did not reveal any significant relationship with the explanatory variables.

**3. Results**

*3.1. Recolonisation of Poisoned Stretches by Sensitive Insect Species*

Eighty-three days after the accident, the sample from non-polluted Site C contained 24 insect taxa and one oligochaete taxon. At polluted Site 1, the species richness was reduced to eight taxa (two oligochaetes and six insects) (Table 3). Site 1 was recolonized by two species classified as SPEAR$_{pesticides}$ (*Ecdyonurus* sp. and *Rhyacophila nubila*) and three species considered as SPEAR$_{refuge}$ (*Ephemera danica*, *Halesus digitatus*, and *Hydropsyche instabilis*) after the poisoning. The two remaining sensitive insect taxa belonged to the Chrironomidae family, represented by an early Chironomidae gen. sp. instar and *Brillia bifida*.

**Table 3.** Summary of macroinvertebrate community parameters calculated for each sample within the sampling period (2014–2017). Abbreviations: Shannon–Wiener index (H′), number of species (N), Abundance (Abu). For meaning of parameters, see Section 2.4.3.

| Parameter | 2014 | | | 2015 | | | 2016 | | | 2017 | | |
|---|---|---|---|---|---|---|---|---|---|---|---|---|
| | C | 1 | 2 | C | 1 | 2 | C | 1 | 2 | C | 1 | 2 |
| Abundance | 1142 | 228 | 2752 | 644 | 966 | 4472 | 2842 | 2313 | 7227 | 3308 | 776 | 1432 |
| Richness | 26 | 9 | 15 | 39 | 26 | 23 | 34 | 22 | 21 | 43 | 23 | 26 |
| H′ | 2.40 | 1.44 | 1.51 | 3.10 | 2.06 | 1.69 | 2.05 | 2.44 | 1.53 | 2.99 | 2.17 | 2.77 |
| SPEAR$_{pesticides}$ index | 0.23 | 0.15 | 0.02 | 0.11 | 0.13 | 0.09 | 0.20 | 0.06 | 0.05 | 0.16 | 0 | 0.08 |
| SPEAR$_{refuge}$ index | 0.25 | 0.29 | 0.05 | 0.14 | 0.15 | 0.02 | 0.06 | 0 | 0 | 0.12 | 0.08 | 0.10 |
| SPEAR$_{pesticides}$ index (old) | 0.48 | 0.44 | 0.07 | 0.25 | 0.29 | 0.11 | 0.26 | 0.06 | 0.05 | 0.27 | 0.08 | 0.18 |
| N SPEAR$_{pesticides}$ | 6 | 2 | 1 | 5 | 4 | 3 | 9 | 3 | 1 | 9 | 0 | 3 |
| N SPEAR$_{refuge}$ | 7 | 3 | 1 | 6 | 5 | 1 | 2 | 0 | 0 | 5 | 2 | 3 |
| N SPEAR$_{pesticides}$ (old) | 13 | 5 | 2 | 11 | 9 | 4 | 11 | 3 | 1 | 14 | 2 | 6 |
| Abu SPEAR$_{pesticides}$ | 306 | 8 | 1 | 40 | 30 | 26 | 94 | 9 | 68 | 136 | 0 | 24 |
| Abu SPEAR$_{refuge}$ | 142 | 44 | 6 | 58 | 36 | 2 | 40 | 0 | 0 | 280 | 16 | 72 |
| Abu SPEAR$_{pesticides}$ (old) | 448 | 52 | 7 | 98 | 66 | 28 | 134 | 9 | 68 | 416 | 16 | 96 |
| N EPT | 14 | 5 | 2 | 12 | 10 | 4 | 7 | 0 | 1 | 11 | 2 | 4 |
| Abu EPT | 448 | 52 | 7 | 148 | 94 | 28 | 100 | 0 | 68 | 396 | 16 | 80 |
| EPT% | 42.73 | 22.81 | 0.25 | 22.98 | 9.73 | 0.63 | 3.52 | 0 | 0.94 | 11.97 | 2.06 | 5.59 |
| Original BMWP score | 100 | 42 | 26 | 107 | 77 | 42 | 102 | 49 | 38 | 141 | 65 | 72 |
| ASPT index ($\pm$SEM) | 6.67 $\pm$ 0.77 | 6.00 $\pm$ 1.35 | 3.71 $\pm$ 0.78 | 6.29 $\pm$ 0.78 | 5.50 $\pm$ 0.79 | 4.20 $\pm$ 0.63 | 5.10 $\pm$ 0.62 | 4.08 $\pm$ 0.53 | 3.80 $\pm$ 0.79 | 5.88 $\pm$ 0.63 | 4.64 $\pm$ 0.70 | 4.80 $\pm$ 0.76 |

At Site 2, 10 Insecta taxa, 3 Oligochaeta taxa and 2 Gastropoda taxa were detected. Two Diptera families (Chironomidae: Chironomidae gen. sp., *Chironomus riparius* gr., *Corynoneura* sp., *Micropsectra* sp., *Paratrichocladius rufiventris*, *Thienemannimyia* sp. and *Tvetenia verralli* and Simuliidae: *Simulium vernum*) dominated the insect assemblage. The remaining two detected insect taxa, both caddisflies classified as SPEAR$_{pesticides}$ (*Rhyacophyla* sp.) and SPEAR$_{refuge}$ (*Hydropsyche* sp.), were detected in very low numbers (one and eight individuals, respectively).

The SPEAR$_{pesticides}$ and SPEAR$_{refuge}$ detected at Site C but not in the poisoned stretches were *Ephemerella mucronata*, *Habrophlebia lauta*, *Serratella ignita*, *Leuctra* sp., *Nemoura* sp., *Hydropsyche siltalai*, *Polycentropus flavomaculatus* and *Potamophylax luctuosus*. The species not at risk (SPEnotAR$_{pesticides}$) that were not found to colonise at least one of the two contaminated Sites were *Microtendipes chloris* gr., *Orthocladius* sp., *Tanypodinae* gen. sp., *Tvetenia verralli*, Ceratopogonidae gen. sp., *Dicranota* sp., *Tipula maxima* and *Elmis* sp. lv.

### 3.2. Relationship between Disturbances and Macroinvertebrate Taxonomic Composition

A Monte-Carlo permutation test performed within independent RDA revealed the significant power of both variables (adjusted $p < 0.05$) for predicting the taxonomic compositions of the whole macroinvertebrate assemblage, which could be classified into seven deeper taxonomic levels (mostly classes) (Figure 2A), and the arthropod assemblage classified into 19 deeper taxonomic groups (mostly orders) (Figure 2B). Sediment contamination by insecticides (CPS-CP) negatively correlated with all the deeper taxonomic levels (in terms of their abundance) other than with Platyhelminthes (*Polycelis nigra*) (Figure 2A). Platyhelminthes, Crustacea, Bivalvia, Gastropoda and Insecta were the most reliably fitted groups. Hence, sensitivity to contamination, from the most to the least affected taxonomic groups, is as follows: Bivalvia (only *Pisidium casertanum* and *P. personatum*) and Gastropoda < Insecta and Crustacea (Ostracoda and Isopoda) < Platyhelminthes (*Polycelis nigra*) (no effect). The remaining two annelid taxa (Hirudinea and Oligochaeta) did not reach the same quality of fit; nevertheless, the Oligochaeta group does seem to be sensitive to CPS-CP contamination. The increasing intensity of drying up positively correlated with densities of all given taxa, but mostly with the Bivalvia species, Crustacea, Hirudinea, and Oligochaeta.

Predominantly lotic Heteroptera and adult Coleoptera were the commonest insect taxa in assemblages found at an advanced stage of drying up (Figure 2B). Conversely, the family Simuliidae was highly sensitive to drought; the EPT group was negatively correlated with both factors. This group showed relatively higher resilience to contamination compared to the other arthropods; however, it was almost completely lacking in samples taken under the worst hydrological conditions (Table 3).

The test performed using the CCA to ordinate the whole assemblage into the highest possible taxonomic level revealed the marginal effect of both predictors (adjusted $p = 0.055$ for the effect of both disturbances).

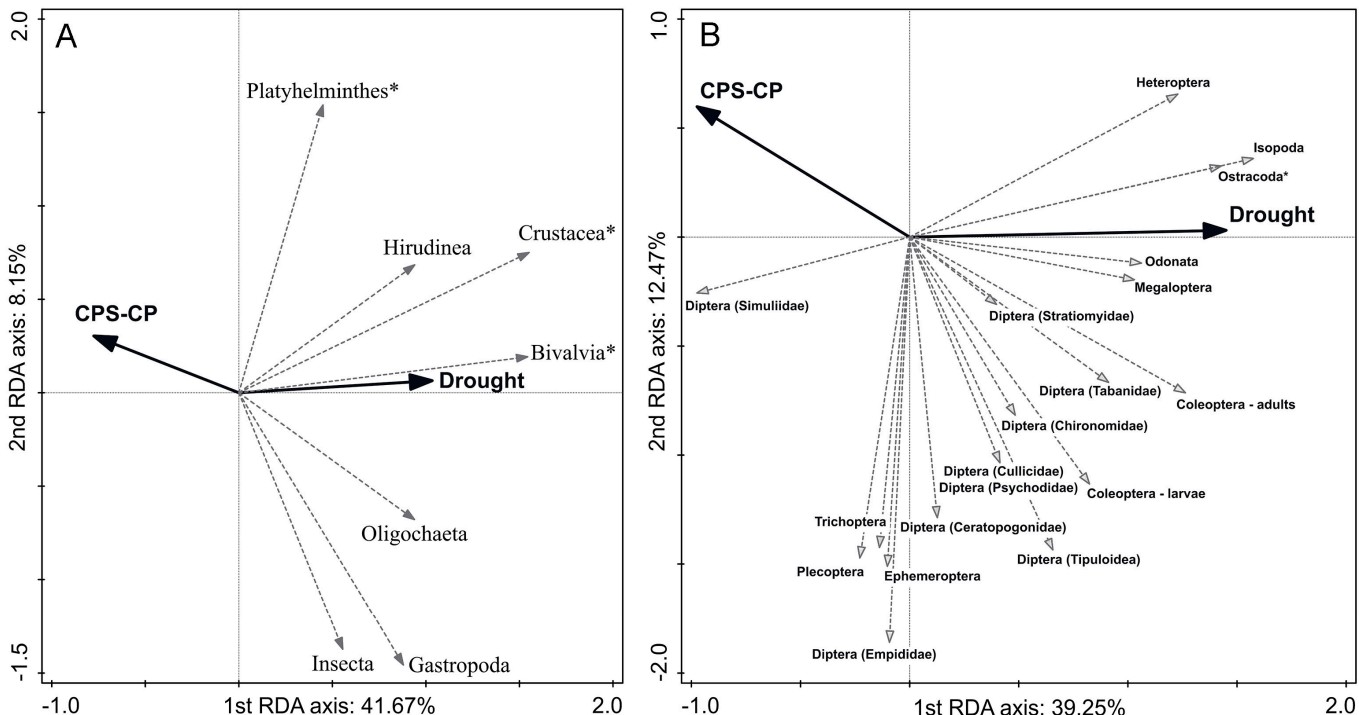

**Figure 2.** RDA ordination diagrams showing the relationships between disturbances (CPS-CP = chlorpyrifos-cypermethrin contamination derived from the chlorpyrifos sediment contamination, and Drought = drought status; see Materials and Methods) as solid black arrows, with taxa represented by grey arrowed lines. The power of the both disturbances to predict taxonomic compositions was significant (adjusted *p* < 0.05). The length of arrows corresponds to the quality of fit. (**A**) Relationships between disturbances and the composition of deeper taxonomic levels extrapolated from the abundances of all detected taxa (CPS-CP: explained variation 26.9%, pseudo-F = 3.3; Drought: explained variation 22.9%, pseudo-F = 3.7, respectively; adjusted variation explained by both predictors = 37.28%). Additional information for the labels marked by the asterisks: Platyhelminthes is represented by a single species—*Polycelis nigra*; Crustacea is represented by Ostracoda (not determined into a deeper taxonomic level) and *Asellus aquaticus*; Bivalvia is represented by only two species—*Pisidium casertanum* and *P. personatum*. (**B**) Relationships between disturbances and the composition of deeper taxonomic levels extrapolated from the abundances of arthropods. (CPS-CP: explained variation 24.7%, pseudo-F = 3.0; Drought: explained variation 27.0%, pseudo-F = 4.5, respectively; adjusted variation explained by both predictors = 39.65%). The asterisk marking Ostracoda indicates that this group was not determined into a deeper taxonomic level.

### 3.3. Effect of Disturbances on Ecological and Biological Species Traits

Of the ecological and plasticity traits described in Materials and Methods (see Section 2.4.2 *Estimation of Species Traits*), only the *Functional feeding groups* for all detected taxa (Figure 3) showed a significant relationship with the first ordination axis and both predictors (adjusted *p* < 0.05) within the partial RDA. The CPS-CP contamination negatively correlated with the abundance of *predators* and *active filterers*, both of which, however, prevailed over the other feeding strategies during drought conditions (Figure 3). This trend was caused by increased densities in, especially, chironomid, ostracod and bivalvian assemblages in isolated pools: namely, the facultative *active filterers* from the Tanytarsini tribe (Chironominae) such as *Micropsetra apposita* and obligatory *active filterers* represented by Ostracoda and Bivalvia (*Pisidium* sp.). Numerous *predators* or facultative *predators* from the Tanypodinae subfamily including *Macropelopia nebulosa*, *Procladius* (*Holotanypus*) sp., and *Thienemannimyia* sp. were followed by the nectonic heteropteran *Notonecta* sp. and the adults of the coleopterans *Platambus maculatus* and *Agabus didymus* during drying up conditions. On the other hand, contamination correlated positively with collector-gatherers for a short time period; nevertheless, this strategy was suppressed during the hydrological droughts, as occurred with passive filtration and grazing-scraping. The shredders

and xylophages were the least involved in material processing during both peaks of the perturbations (Table S3).

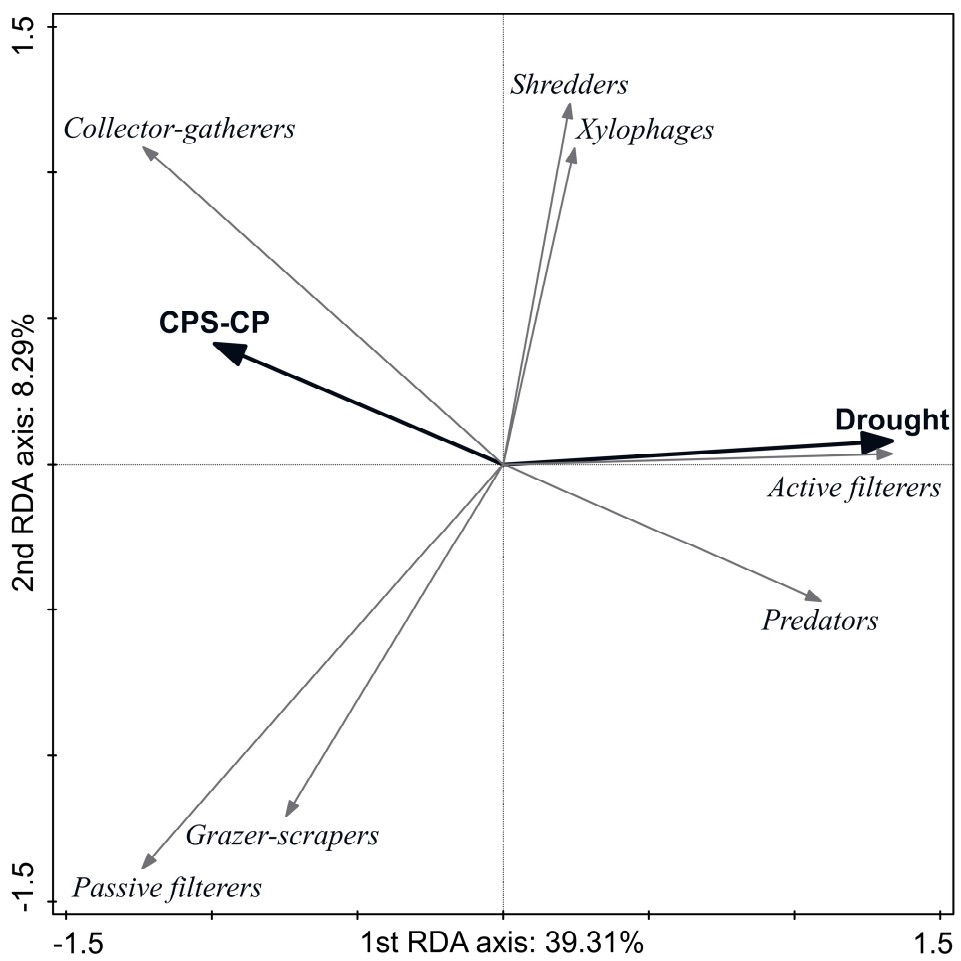

**Figure 3.** Ordination diagram showing the relationships between the average functional feeding groups (FFGs) represented by solid grey arrows and disturbances (CPS-CP = chlorpyrifos-cypermethrin contamination estimated from the chlorpyrifos sediment contamination and Drought = drought status; see Materials and Methods) represented by solid black arrows. Both disturbances have significant ($p < 0.05$) power to predict the FFG composition. Arrow length corresponds to the quality of fit. The FFG was extrapolated from abundances of all detected organisms (CPS-CP: explained variation 25.3%, pseudo-F = 3.1; Drought: explained variation 22.3%, pseudo-F = 3.4, respectively; adjusted variation explained by both predictors = 34.50%).

The remaining compositions of species traits extrapolated from taxa abundance did not significantly correspond with the variables. Nevertheless, relationships between disturbances and compositions of species traits (listed in the Section 2.4.2 *Estimation of Species Traits*) are shown in the diagrams resulting from the PCA (Figure S1).

## 4. Discussion

### 4.1. Response of Macroinvertebrate Taxa to the Disturbances

Macroinvertebrate species assemblages can substantially change after acute poisoning with insecticides and during prolonged periods of drought [53,66,67]. The significant relationship plotted in Figure 2A suggests that no groups of organisms prefer the CPS-CP contamination, and that only the Platyhelminthes (*Polycelis nigra*) tolerate it. Contrary to our prediction (Table S1) that the stream macroinvertebrate assemblage would shift from being dominated by arthropods to being dominated by non-arthropod species, the insect and crustacean groups, in general, did not seem to be the most affected groups [67]. The most likely cause of this effect is that the downstream part of the brook was colonized

by species from the upper non-affected part. In addition, the insect species with the ability to drift downstream or with winged imagines (ephemeropterans, plecopterans, and chironomids) are often perfectly adapted for colonizing new habitats, and so insects may in fact be very resilient to insecticides [34]. These adaptations may be much more advantageous than a resistance to a prolonged or severe disturbance [68,69]. However, if species populations are too weak or non-existing in adjacent brooks or catchments, they can become temporarily or permanently extinct [70].

Hydrological conditions—and especially droughts—heavily influenced the macroinvertebrate assemblage. The abundance of all groups correlated more or less positively with an increase in droughts (Figure 2A). Bivalves (*P. casertanum* and *P. personatum*), gastropods, oligochaetes, crustaceans (*Asellus aquaticus* and Ostracoda), as well as leeches, all showed greater correlation than insects (bivalves and gastropods may have increased their relative density due to a decrease in available space during droughts). Furthermore, the drought period was accompanied by increased sedimentation of organic matter caused by very low water discharge. A further meaningful biological factor that could be partially responsible for both the redundant accumulation of organic matter and assemblage shift is the total extinction of *A. astacus* in this brook. Before the accident, noble crayfish were found at a density of around one individual per m$^2$ at Sites 1 and 2, and its population did not recover at any point during this study (no individuals caught in traps) [71]. These bottom-dwellers shred and release leaf litter from under obstacles on riverbed into the current, dig burrows in riverbanks, and also feed on macrozoobenthos, presumably slow-movable macroinvertebrates [72].

The Crustacea group represented by the isopod *A. aquaticus* and the class Ostracoda (Figure 2A) appeared at all Sites during the dry period in 2016–2017, as did the Bivalvia. The relationship between CPS-CP and this group was similar to the relationship between CPS-CP and the insect group, which demonstrates the high specific toxicity of these compounds to arthropods in general [73,74]. Nevertheless, the drought had a generally positive effect on these two crustaceans (Figure 2A,B). The relationship between insecticide pollution and the abundance of *A. aquaticus* within ecosystems that are often affected by toxicants as well as organic pollution remains unclear [40]. Although *A. aquaticus* is very sensitive to acute exposure to insecticide, it can tolerate lower concentrations of various toxic compounds, including insecticides [75–77]. It is also able to avoid desiccation [78]; furthermore, Extence [79] reported a significant increase in the abundance of *A. aquaticus* in a lowland river during a drought period as a result of greater accumulation of organic matter. The response of *A. aquaticus* to these two types of disturbances can be summed as intolerance to acute poisoning but a general preference for an ecosystem disturbed by partial drying up and the increased deposition of organic material that can serve as a food resource.

The relationships between arthropod groups and disturbances shown in Figure 2B suggests that very few taxa were positively correlated with the CPS-CP contamination; however, the individual response to greater hydrological drought was more complex. Members of the family Simuliidae were the most negatively correlated with greater drought (and had a positive relationship with CPS-CP contamination), a logical finding given that, due to their feeding adaptations, they are dependent on flowing water current [80]. We believe that the positive relationship with increasing CPS-CP contamination is due to their ability to recolonize contaminated stretches of brook if favourable hydrological conditions are present (i.e., flowing water) [81]. The general intolerance of many species from the EPT group to pesticides, organic pollution, and a lack of water (factors that are typical in water courses affected by large-scale agriculture) has been well documented and is discussed in more detail below (Section 4.2) [82–84].

Taxonomic groups ordinated near the Chironomidae family were represented by the larvae of other dipteran species and coleopteran larvae and adults. This group of taxa seems to be very sensitive to acute pollution; however, it colonised the most contaminated Sites immediately after the mayflies and caddisflies and, unlike them, were tolerant to

drought. This also occurred in the families such as the Dytiscidae that colonising shrinking pools. Chironomids are the most complex family and individual species have very plastic ecological characteristics [85–88]. Their high susceptibility to CPS-CP and other insecticides under laboratory conditions does not correspond to the generally high resilience of many Chironomid species to various types of disturbances [89,90]. However, the preferences of chironomid larvae for certain hydrological conditions varies between species within subfamilies or even within a single genus. For instance, the subfamily Orthocladiinae decreased in abundance and richness during the dry period, whilst the Tanypodinae and Tanytarsini (*Micropsectra* sp.) dominated in the samples during drought conditions in 2016 (Table S2). The high densities of *Micropsectra* sp. could be a product of the stress caused by the drought, and numerous tanypodins would either feed on them or be similarly forced to move into refuges [84,91]. It is worth mentioning the fact that stream macroinvertebrates often concentrate in small wet areas during periods of drought, and that they can increase or decrease their densities depending on taxon-specific life histories [54]. This system may be much more at risk to different types of pollution and other interferences, and higher densities and weakening may favor the spread of disease within assemblages [92].

### 4.2. Response of Species-Trait Assemblages to Disturbances

Taxonomic and functional feeding group (FFG) assemblages often have intimately interrelated characteristics that change along environmental gradients and may react in different ways to disturbances [54]. Less time is usually required for the recovery of trophic structures than for the recovery of taxonomic assemblage structures [93]. Although shredders are considered to be the most susceptible group to insecticides, and despite the fact that they are often also sensitive to droughts, the approximal correlation between shredders and both drought and CPS-CP was close to zero [53,67]. Almost identical trends were estimated for xylophages. The relative abundance of shredders and xylophages was low at all Sites (average proportion to other FFGs: 6.96% and 1.39%, respectively), whilst collector-gatherers dominated in most cases (average proportion to other FFGs: 50.01%; Table S3). This could have been caused by the time of sampling (April–September) because shredder abundance commonly increases in autumn when more leaf litter is available [94]. However, if the whole assemblage were chronically influenced by insecticides at all Sites, an arthropod-shredder population, represented especially by caddisfly larvae (no gammarids were detected during the study), would be suppressed. Specifically, the shredder FFG was represented by the caddisfly *H. digitatus*, which prefers permanent waters, at the contaminated Sites after the accident, whilst the isopod *A. aquaticus*, which is considerably more tolerant to drought conditions, was the most abundant representative of the facultative shredding strategy during the period when the brook was drying up [78,95]. The exclusion of shredders from headwaters is thought to trigger ecological changes in overlapping assemblages due to the crucial role they play in the smooth cycling of nutrients by processing coarse particulate organic matter into smaller transportable particles (fecal pellets and orts). The headwater streams lacking shredders are colonized by saprotrophic microorganisms (bacteria, fungi, and protozoa), followed by microphagous collector-gatherers and filterer macroinvertebrates. This type of assemblage is not very resistant to the highly turbulent water flow found in headwaters after heavy precipitation [25,67,96,97].

The positive relationship between drying up and predator abundance has been described by Boulton and Lake [53], who named this phenomenon as 'predator soup'. It occurs as organism density increases in shrinking pools. Herbst et al. [98] reported an expansion of micro-predators (e.g., Tanypodinae, and Ceratopogonidae) that was partially the case in our study. The possible proliferation of an active filterer FFG assemblage in temporary standing water (e.g., Culicidae) was also suggested [53]. Nevertheless, a correct interpretation is more difficult because part of the active filtering organisms, e.g., Tanytarsini chironomids, would be forced to move into residual pools with negative consequences for their survival [84,91]. Passive filters rely on water currents; thus, this feeding strategy may be useless when stream flow ceases. Nevertheless, within a species

this strategy is usually combined with other options for getting food. Similarly, the grazer-scrapers, being the second most dominant FFG after collector-gatherers, were negatively correlated with the drought variable. Despite the fact that this strategy could co-exist with others in a single species, many studies describe it as existing on its own without any other feeding mechanism, e.g., in certain heptagenid nymphs, gastropods, etc. [99]. Herbst et al. [98] reported a decrease in the relative abundance of grazers during a severe drought. According to our results, the abundance of grazers correlated negatively with the Drought variable, while its correlation with CPS-CP was close to zero.

### 4.3. Parameters of a Macroinvertebrate Assemblage at the Studied Localities

Stressor-specific eco-indicators are a useful tool for monitoring ecosystems because they reveal ecological effects and assess how single and combined stressors affect ecosystem structure and function [40,50,100]. The parameters calculated for each sampling Site (Table 3), which showed an obvious shift in a macroinvertebrate assemblage between localities, were not statistically supported. Although the multihabitat-sampling method used to assess the macrozoobenthos status is robust [62], this approach is often inadequate for a standard research analysis. However, in this particular study, the dataset was of use due to the very strong effect of the studied ecological disturbances. It is hard to interpret differences in total abundances from the data; nevertheless, fivefold lower macroinvertebrate abundance 2.5 months after poisoning in 2014 at Site 1 compared to Site C does suggest that macroinvertebrate density did not recover. Cuffney et al. [67] propose that approximately four months are needed for macroinvertebrate density to recover after an acute poisoning event (even though the biomass will remain significantly lower in the first year before recovering in the second [101]). However, in these authors' study the recolonization process from an upstream section did not occur, unlike in our study. The richness, BMWP score and ASPT index estimated for Site C were regularly higher than at Sites 1 and 2, which reveals the negative impact at both Sites during the monitoring period. The BMWP approach, which considers only the richness of specific family (not abundance), indicates the presence of a higher level of saprobity at these Sites [64].

Insecticide pollution chiefly affects the most susceptible stream insect assemblages [67]. The abundance (Abu) and richness (N) of the indicator species (SPEAR and EPT) decreased at the contaminated Sites; however, the indices calculated from the proportion of the sum of the sensitive species abundances vs. the sum of all species abundance (even if log-transformed) are intended for use in cases of chronical stress, not for events of massive spills of toxicants leading to the total extermination of all species [40]. Therefore, the conclusions resulting from these indices could be misinterpreted. For example, when comparing Site C with Site 1 in 2014, both samples had almost the same value for the SPEAR$_{pesticide}$ index (old), despite the huge contamination that had occurred there!

In terms of assemblage parameters (richness, H', SPEAR, EPT-derived values; see Table 3), site C was the least disturbed environment—despite its similar or even worse hydrological situation—when compared to Sites 1 and 2 in 2016 and 2017. Nevertheless, the values for the SPEAR$_{pesticides}$ index calculated for each sample, including these from Site C, reflect its 'poor' or 'bad' ecological status (according to the EU Water Framework Directive) [49,102]. These findings could be explained by the lack of any well-conserved refuge area. Sufficiently long stretches of uncontaminated headwaters situated in riparian forests are thought to act as significant functional refuges enhancing stream hydro-morphology and enabling the recolonization of a damaged downstream section [103]. However, pollution by an insecticide or any other kind of pesticides can often occur even in economically exploited forests [104].

Many authors consider that 'edge-of-field' runoff after heavy rain represents the main vector for pesticides entering water ecosystems [9,40,105,106]. According to Stehle and Schulz [52], the strength of their effect can be modulated by application patterns, geographical and meteorological conditions, the physicochemical properties of the insecticide (e.g., stronger sorption properties can reduce the impact) and its intrinsic toxicity. The basin of

the Doubravka brook including its source is exploited agriculturally. The excessive size of local fields (up to 100 ha) drained by surface and sometimes by sub-surface systems, combined with steep slopes, undoubtedly increases the risk of pollution by agricultural chemicals, of extreme hydrological situations and of soil erosion [44,45].

Immediately after the accident (2014), total invertebrate abundance and richness were substantially lower near the contamination source. The first pioneers to colonize Site 1 were the SPEAR$_{pesticides}$ (old) and EPT species, although there were also a number of SPEAR$_{refuge}$. Despite their high sensitivity to pesticides, these species are frequently detected in contaminated stretches near uncontaminated stream sections regardless of the pesticide pressure. The occurrence of SPEAR$_{refuge}$ in contaminated or disturbed streams can be caused by dispersal-based resilience and the ability of these organisms to disperse from uncontaminated to contaminated stream sections (via drift and adult dispersal) [49]. Given the spatial proximity between uncontaminated Site C situated upstream from Site 1 (Figure 1) and the time lapse between the accident and the sampling time (2.5 months), the drift of these insect species is the most likely explanation of their occurrence there. In addition, the proportion of stream organisms drifting downstream from stretches damaged by repeated insecticide pollution can be much higher than in undamaged stretches [67]. The situation at all sampling Sites (low values of SPEAR$_{pesticide}$ indices, agricultural exploitation of the basin, and the period of pesticide application between the accident and the first sampling) suggests chronic pollution by pesticides or another disturbance. Nevertheless, the lethality of short-term pollution and the level of macroinvertebrate drift from Site C is unknown; thus, recolonization via drift downstream from this Site could either be accelerated by a lower-effect disturbance, especially at the level of sensitive species richness, or inhibited by a higher-effect disturbance at this level and, especially, at a level of sensitive species abundance. It is worth mentioning that fresh generations of chironomids and Oligochaeta (the Naidinae subfamily) were detected at Site 1, although both groups are considered to be SPEnotAR$_{pesticides}$, and they could even have been reproducing at this Site after the accident. Since the bottom of contaminated stretches of brook became covered by large amounts of filamentous algae and periphyton during the first few months after the accident, these organisms could have prospered in this micro-habitat despite the persistence of the contamination in sediments (Figure S1B).

The distance between sampling Site C and sampling Site 2 (approximately 6 km) could represent the threshold distance needed for fast recolonisation by SPEAR$_{refuge}$ species [49]. This agrees with our findings because the detected abundance and richness of SPEAR$_{pesticides}$ and SPEAR$_{refuge}$ species were very low here compared to Site 1, although Site 2 was not damaged as much by the gradual flow of contaminated sediment from the drainage channel. On the other hand, the sensitive insect species (SPEnotAR$_{pesticides}$) represented by Diptera (Chironomidae and Simuliidae) might have recolonized this stretch given their reproductive potential. Since the detected dipteran larvae were mostly bi- or multi-voltine (and in a case of later instars, above all epibenthic), survival in the hyporheic zone is less likely. In the case of caddisflies, migration from the upstream section and/or from a possible small tributary refuge situated between Sites 1 and 2 cannot be excluded.

In 2016 and 2017, the almost total absence of SPEAR$_{pesticides}$ and EPT species at Sites 1 and 2 (Table 3) raises several possible questions about another pesticide contamination event (or another disturbance) during this period. This event was partially revealed by the increased concentration of cypermethrin recorded in sediments sampled in the drainage upstream from the confluence with the Doubravka brook. Despite this, a higher concentration of cypermethrin was not detected at the observed Sites. During this period, the area was affected by severe hydrological drought. Given that the drainage channel during the time period supplemented Sites 1 and 2 with muddy, pesticide-enriched water, this situation is a good example of the combined effects of drought, greater sedimentation and pesticide pollution ('ramp', 'press', and 'pulse' types of disturbance, respectively, according to Lake [107]), which led to partial absences of the EPT species, especially at Site 1 (Table 3). Despite the fact that locality C was affected by the same or even greater lack of

water than these two localities (Table S5), the EPT groups resisted here. Nevertheless, the redundant sedimentation caused by the lack of water current probably triggered the high development of chironomids and oligochaetes.

## 5. Conclusions

This case study attempts to describe how the macroinvertebrate assemblage responds to acute poisoning by insecticides (Chlorpyrifos and Cypermethrin) and severe hydrological droughts in a headwater stream that is chronically affected by agrochemicals and large-scale agriculture exploitation in its catchment area. In the case of the accidental insecticide contamination, the assemblage reacted with a drop in richness and abundance. The contaminated Sites were colonized by resilient taxa and the lotic insecticide-susceptible taxa were able to spread from non-contaminated refugial stretches situated upstream from the source of the contamination. In the case of hydrological droughts, lotic organisms declined and the assemblage shifted to favor taxa that prefer higher levels of organic pollution and lentic hypo-neustonic (nektonic) predatory insects.

A non-stable hydrological regime is a serious issue that many once-permanent aquatic ecosystems have to confront. Damage caused by agricultural pollution will be more serious for sensitive taxa if headwater ecosystems turn into semi-temporary systems. Nevertheless, the consequences of droughts are of greater concern to ecologists and administrative bodies. We are convinced that the rationale management of headwater basins is key in ensuring the proper ecological status of whole river networks, which are essential for human society in many aspects.

**Supplementary Materials:** The following are available online at https://www.mdpi.com/article/10.3390/w13101352/s1, Table S1: Presumed negative effect of chlorpyrifos and cypermethrin on chosen invertebrate taxonomic groups; Table S2: Abundance and richness of chironomid subfamilies and tribes in individual samples; Table S3: Percentage proportion of functional feeding groups in terms of abundance in individual samples; Figure S1: PCA ordination diagrams showing relationships within the composition of given taxa traits and between these traits and particular explanatory variables; Table S4: Meteorological data of the Vysočina Region (Czech Republic), containing the catchment area of the Doubravka brook, for individual years of the invertebrate sampling; Figure S2: The hydrologic situation during the sampling campaign 2014–2017 in the Doubrava river (Spačice, Czech Republic); Figure S3: Photographic documentation of different hydrologic situations through the year 2016 at the Site C (control = unaffected, Transect 1); Figure S4: Photographic documentation of different hydrologic situations through the year 2016 at the Site 1 (affected, Transect 1); Figure S5. Photographic documentation of different hydrologic situations through the year 2016 at the Site 2 (affected, Transect 2); Table S5: The hydrological status score assigned to each sample during the sampling period 2014–2017; Table S6: Detected taxa and their semi-quantitative abundances in all samples; Table S7: Detected taxa and their ecological and plasticity traits.

**Author Contributions:** Conceptualization, J.Š., A.K., and M.B.; methodology, J.Š., M.L., and M.F.; formal analysis, J.Š. and M.L.; writing—original draft preparation, M.L. and M.B.; writing—review and editing, A.K., M.B., and J.Š.; All authors have read and agreed to the published version of the manuscript.

**Funding:** This research was funded by the Ministry of Agriculture of the Czech Republic, project no. QK1910282.

**Institutional Review Board Statement:** Not applicable.

**Informed Consent Statement:** Not applicable.

**Data Availability Statement:** The data presented in this study are available in the supplementary material.

**Acknowledgments:** We would like to thank Lukáš Veselý for valuable advices and Miloš Buřič and Jan Kubec for their contribution during field works.

**Conflicts of Interest:** The authors declare no conflict of interest.

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
