# Peer review of "Insecticides and Drought as a Fatal Combination for a Stream Macroinvertebrate Assemblage in a Catchment Area Exploited by Large-Scale Agriculture"

_water, doi:10.3390/w13101352_

Round 1

Reviewer 1 Report

There is a marked difference in the writing style of the introduction and the rest of the manuscript. The introduction will benefit from an in-depth review and background information. Substantial English editing needs to be done in the introductory section.

Detailed comments and edits:

Line 37: remove “sometimes”

Line 37-40: This statement contradicts the statements in the previous sentences.

Line 44: Exchange “fails to respect” with “intervenes”

Line 45-47: The sentence is confusing. Consider changing it to “Affected headwaters will dry up more often and more quickly, which will exacerbate the effects of pollution by point and non-point sources.

Line 48: it is not only xenobiotics, many different types of pollutants. Exchange “xenobiotics” with “pollutants”

Line 55: Exchange sentence “Therefore, pesticides or their residuals can periodically have acute or chronic effects on aquatic organisms [41,42], and toxicants with great affinity to soil or organic matter may persist in sediments [18,43].” with “Due to their high affinity to soil or organic matter, pesticides can persist in sediments. Exposure to pesticides can result in acute or chronic effects in aquatic organisms.“

Line 60: I cannot agree that the presence of pesticides results in an increased occurrence of hydrologic events!!!

Line 68: exchange “whole” with “entire”

Line 69: Use “group” not “groups”.

Line 72: Exchange “experiments carried out in natural scenarios” with “large-scale experiment” OR “field experiments”

Line 91: exchange “brook” with “stream”

Line 91: remove “and”

Line 92: exchange “rises” with “starts”

Line 93: exchange “watercourse” with “stream”. I cannot agree that a fourth-order stream will become a third-order as it flows downstream. It is vice-versa.

Line 116: exchange “poisoned” with “contaminated”

Line 187: Report the range of the % recoveries of the spiked samples

Line 208: capitalize the word “Site”.

Line 213: Remove “primary purpose of the”. Add “was initiated” after the word “sampling”

Line 422: remove extra space between “the” and “high”

Author Response

Dear reviewer,

We are submitting our revised manuscript “Insecticides and drought as a fatal combination for a stream macroinvertebrate community in a catchment area exploited by large-scale agriculture".

In this document below, please find the point-by-point answers to all constructive comments and revisions. We tried to fulfill all requirements and suggestions to improve the manuscript and increase its scientific quality. We respond to all suggestions and requirements. Concurrently, we added our own improvements (page n. 7 of this document).

Please, find the manuscript and supplement files with modifications visible by track changes (inside a zip folder “Let et al., 2021 revised”: word files – “Let et al.,2021 revised” and “Let et al.,2021 supplement files revised”, respectively). In addition, we attached updated figures inside in a separate folder – “Let et al.,2021 figures updated”.

Finally, we want to emphasize that although language editing was recommended, our manuscript has been proofread in British English by a native speaker, Dr. Michael Lockwood (currently working in Catalonia on bird, butterfly and dragonfly studies) who makes English corrections of manuscripts for our department.

Thank you for your time and effort when evaluating our revised contribution.

Yours faithfully,

Marek Let (on behalf of collective authors)

Response to Reviewer 1 Comments

Line 37: remove “sometimes”

Our response: We changed accordingly (line 40). We admit that omitting “sometimes” here improves the stylistic form of the text.

Line 37-40: This statement contradicts the statements in the previous sentences.

Our response: We changed the sentences to be more understandable (lines 36–43). We intended to highlight that human activities are not always linked to only negative effects (lines 39–43), which is worth of mentioning.

“Macroinvertebrate communities inhabiting headwaters in a cultural landscape have to cope with exposure to anthropogenic activities and their implications [1-5]. The vast majority of published studies generally associate human activities with negative impacts on stream invertebrate biota and their ecological functioning, that is, they provoke a fall in diversity, abundance, biomass and organic matter processing [6-12]. Nevertheless, human-triggered effects can result in higher biomass of specific aquatic organisms [13-15] or, occasionally, in higher species richness [16,17], and some ostensibly heavily affected sites may sometimes even act as refuges for endangered species [18-20]”.

The changes implicated reordering of cited literature in the references part (see the lines 591–645)

Line 44: Exchange “fails to respect” with “intervenes”

Our response: We changed accordingly (line 49).

“Combined with intensive land use that intervenes functioning water regimes, climate change may cause huge ecological and economic damage [25-28]”.

Line 45-47: The sentence is confusing. Consider changing it to “Affected headwaters will dry up more often and more quickly, which will exacerbate the effects of pollution by point and non-point sources.

Our response: We changed the sentences to be more understandable (lines 50–51).

“Affected headwaters, will dry up more often and more quickly, thereby multiplying the effects of pollution by agricultural or urban wastewater [29-31]”.

Line 48: it is not only xenobiotics, many different types of pollutants. Exchange “xenobiotics” with “pollutants”

Our response: We changed accordingly (line 53).

Line 55: Exchange sentence “Therefore, pesticides or their residuals can periodically have acute or chronic effects on aquatic organisms [41,42], and toxicants with great affinity to soil or organic matter may persist in sediments [18,43].” with “Due to their high affinity to soil or organic matter, pesticides can persist in sediments. Exposure to pesticides can result in acute or chronic effects in aquatic organisms.“

Our response: We cannot see any reason for this change. The original sentence (line 58–60) is linked with the previous two sentences thus we consider the “therefore” on the beginning as adequate and appropriate:

“The main reason is their massive use worldwide and their typically high specific toxicity to non-target aquatic organisms [5,36-38]. Pesticides are an integral part of conventional agriculture and enter the water at much higher concentrations after short-term runoff events than those that are usually detected by standard monitoring sampling methods [39,40]. Therefore, pesticides or their residuals can periodically have acute or chronic effects on aquatic organisms [41,42], and toxicants with great affinity to soil or organic matter may persist in sediments [18,43]”.

Line 60: I cannot agree that the presence of pesticides results in an increased occurrence of hydrologic events!!!

Our response: We used the verb “to accompany” (line 61). According to our knowledge, it has a different meaning than “ to result”; in other words, correlation does not imply causation. The extreme hydrological events are theoretically more frequent due to the human-accelerated outflow which is caused by channelisation and drainage of small watercourses in an agricultural landscape and urban landscape as well. Concurrently, these human-impacted landscapes are usually exposed to pesticides more than the natural ones.

“Their presence in aquatic ecosystems is usually accompanied by a wide range of anthropogenic influences, e.g. habitat degradation, drainage [44,45], artificial siltation and sedimentation [46,47], increased nutrient input, and the occurrence of more frequent extreme hydrological events (droughts and floods) at interconnected sites [44,48]”.

Line 68: exchange “whole” with “entire”

Our response: We changed accordingly (line 70).

Line 69: Use “group” not “groups”.

Our response: We added three examples of indicator groups thus there is no need for the singular form (line 71).

Line 72: Exchange “experiments carried out in natural scenarios” with “large-scale experiment” OR “field experiments”

Our response: We added “natural experiments” in parenthesis (line 74–75) because we based on the terminology of Diamond (1983) [58] who is frequently cited. He divided ecological experiments into three basic groups – “laboratory”, “field” and “natural”, whereas the “natural experiment” has no treatment from the side of researchers. On the contrary, the “field experiment” includes treatment or manipulated settings (e.g., intentional controlled poisoning of stream etc.).

Line 91: exchange “brook” with “stream”

Our response: We changed accordingly (line 89).

Line 91: remove “and”

Our response: We changed accordingly (line 90).

Line 92: exchange “rises” with “starts”

Our response: We do not think it is needed – the “rises” serves well here (=”its spring is located at an altitude of 460 m a.s.l ”) (line 91).

Line 93: exchange “watercourse” with “stream”. I cannot agree that a fourth-order stream will become a third-order as it flows downstream. It is vice-versa.

Our response: We changed accordingly (line 91). For stream order, we used so-called topological ordering. It is valid (see figure 1 below). We added this information to the text (line 90).

Figure 1: Type of stream ordering (https://grass.osgeo.org/grass78/manuals/addons/r.stream.order.html)

Line 116: exchange “poisoned” with “contaminated”

Our response: We changed accordingly (line 112).

Line 187: Report the range of the % recoveries of the spiked samples

Our response: Information was added (line 176–177)

Line 208: capitalize the word “Site”

Our response: We changed accordingly (in the cases where the “Site” represents the name of particular sampling site (lines 206–554).

The capital letter at “Size” was implemented also into supplementary materials (titles of Tables S2 and S3).

Line 213: Remove “the primary purpose of the”. Add “was initiated” after the word “sampling”

Our response: We changed accordingly (line 205).

New modified improvements

Line 14: We changed “response” to plural “responses”.

Line 70–71: We added examples of sensitive indicator species and higher taxonomic groups as a reaction to the recommendation of reviewer #2 with a particular citation (for more see our response above).

Line 98: We removed “and”.

Line 241: …“the organic pollution“ – “the“ was removed.

Line 250: “Fast flowing” was replaced for “Fast-flowing”.

Line 351 (Table 3): The title was enhanced with explanations: “C (unaffected, transect 1), 1 (affected, transect 1) and 2 (affected, transect 2)” but font size had to be decreased from 9 to 8 because of limited space.

Line 360: We added a comma before “in general”.

Line 362: We replaced “In addition” for “Besides”.

Line 528: We replaced “despite the fact that” for “although”.

Line 592–832: We adjusted both, the part “References” as well as the order of cited literature in square brackets inside the main text (line 39–543)

Reviewer 2 Report

This is an interesting study investigating the response of a stream to an accidental insecticide with a follow up which also investigated he impact of drought conditions on benthic communities.  The main body of the paper relies primarily on the results of multivariate analyses to summarize responses over time and distance with two figures and table 3 which presents summary statistics.  The authors have a wealth of data on taxa abundance and composition and refer to these in some depth in the discussion to describe study findings.  My main disappointment with the paper is that I would have liked to have seen more of the actual abundance data graphed for total abundance and select species. I also would have liked to have seen the data demonstrating drought and changes in steam hydrology.  The authors know their subject well with discussions of colonization from upstream sources, losses of crayfish predators, changes in periphyton, etc. but a more visual presentation of the data would more rapidly demonstrate change.  If space is a consideration, such graphs could appear in supporting. I recommend acceptance with minor revisions. A few points are below.

The introduction is well written and introduced the topic well. However, I recommend moving the details of the insecticide contamination event into the introduction as lines 116-130 are more introductory material.  I also was confused by line 127 which cites results (Table 1) which were then followed by five paragraphs on how sediments were analyzed for pesticides.  This whole section is confusing because no information is provided on how the sediments were collected.

What are the acute and chronic toxicity values for CPS or sediment quality guidelines?  

The methods should begin with the current locality description, the go on to sampling periods, sediment collections, macrobenthos collections, lab analyses (sediments and benthos), and data treatment.

The study was conducted in April and June 2014, April 2015, July 2016 and September 2017.  Would some differences across years be affected by seasonality, i.e., spring versus fall sampling?

It does not look like water flow or depth were measured during the sampling intervals or substrates characterized including periphyton and filamentous algae quantities. Drought is inferred as an influencing factor, but climate data are not presented.  

How was pesticide loss from site 1? Biodegradation or flushing of contaminated sediments? The rapid decline at site 1 in two months with littles change at site 2 suggests degradation.  

Lines 259-266 reference climate and hydrological measurements but these sections could occur earlier as part of field measurements. 

Lines 225-226.  I am not familiar with all the terms and some such as xylal, akal, pelal etc, could have a word in brackets to describe as could FPOM which I assume is fine organic matter.

Results section should begin by discussing table 1 which was cited briefly in materials and methods.  Hydrological and climate data also should appear here to be followed by the benthos data.  The hydrological scores for the three sites and 5 sampling times should be shown.

It would have been nice to have seen some actual data for species abundance over time and location.

The control sites appear not to have been affected by the 2016 and 2017 drought with high abundance and richness during these years.  I assume this was a deeper site or with faster flowing water?

The discussion is good, but I would have liked to see taxonomic data by time and location in addition to the multivariate analytical outcomes.  The dry period is not shown in any way in terms of water level, flow, precipitation and so hard to visualize.  All sorts of changes and responses are discussed over time and location but again data are not shown beyond table 2.

Line 518. I do not see a Table 4. Also, lines 604.

Overall an excellent study.  

Reviewer 3 Report

The manuscript water-1161755 entitled “Insecticides and drought as a fatal combination for a stream macroinvertebrate community in a catchment area exploited by large-scale agriculture” aimed to assess the effects of insecticide contamination in a macroinvertebrate community, observing the ability of species to recolonize stretches of the damaged water course. 
The purpose of this study is very interesting and fits well with the aims and scope of Water.
The introduction briefly describes the current state of the research the study in a broad context and define the purpose of the work and its significance, including specific hypotheses being tested. However, I think that some sentences about the macroinvertebrates as indicators of chemical pollution should be added in this section. 
Material and Methods are written with sufficient detail to allow others to replicate and build on published results. Results are well presented. Discussion is well written. Conclusions are supported by the results. I recommend that you have your manuscript professionally edited by an English native speakers.

Specific comments

L. 208. site -->Site
L. 244. Please, write the formula using “equation” in Word
L 261. Please, briefly describe the scoring system reporting the citation as well.
L. 270. Please, indicate the software used to perform the statistical analysis
Line 389-390. These sentences should be moved to the introduction section. Also, some sentences about macroinvertebrates as indicators of chemical pollution and how FFG can influence contaminants accumulation should be added in the introduction (i.e., https://doi.org/10.1016/j.scitotenv.2019.134282; https://doi.org/10.3390/biology9090288). 
L. 618. Conclusion section should include few sentences. Please, reduce the section accordingly (only the key points should be reported). 

Author Response

Dear reviewer,

We are submitting our revised manuscript “Insecticides and drought as a fatal combination for a stream macroinvertebrate community in a catchment area exploited by large-scale agriculture".

In this document below, please find the point-by-point answers to all constructive comments and revisions. We tried to fulfill all requirements and suggestions to improve the manuscript and increase its scientific quality. We respond to all suggestions and requirements Concurrently, we added our own improvements (page n. 4 of this document).

Please, find the manuscript and supplement files with modifications visible by track changes (inside a zip folder “Let et al., 2021 revised”: word files – “Let et al.,2021 revised” and “Let et al.,2021 supplement files revised”, respectively). In addition, we attached updated figures inside in a separate folder – “Let et al.,2021 figures updated”.

Finally, we want to emphasize that although language editing was recommended, our manuscript has been proofread in British English by a native speaker, Dr. Michael Lockwood (currently working in Catalonia on bird, butterfly and dragonfly studies) who makes English corrections of manuscripts for our department.

Thank you for your time and effort when evaluating our revised contribution.

Yours faithfully,

Marek Let (on behalf of collective authors)

Response to Reviewer 3 Comments

  1. 208. site -->Site

Our response: We changed accordingly (in the cases where the “Site” represents the name of particular sampling site (lines 206–554).

The capital letter at “Size” was implemented also into supplementary materials (titles of Tables S2 and S3).

  1. 244. Please, write the formula using “equation” in Word

Our response: We changed accordingly (line 230).

L 261. Please, briefly describe the scoring system reporting the citation as well.

Our response: We added “semi-quantitative” (line 244).

This scoring system follows a standard approach how to quantified rather categorical factors, and due to the non-parametric approach of the Monte-Carlo permutation method and it is considered to be usable for testing of hypotheses in ecology (for more information, see e.g., Šmilauer and Lepš, 2014). A similar but not identical approach was used by Aspin et al. (2019). However, we used levels of drought intensity described by Boulton and Lake (2008).

Šmilauer, P., & Lepš, J. Multivariate analysis of ecological data using CANOCO 5. Cambridge university press 2014.

Aspin, T. W., Hart, K., Khamis, K., Milner, A. M., O'Callaghan, M. J., Trimmer, M., ... & Ledger, M. E. Drought intensification alters the composition, body size, and trophic structure of invertebrate assemblages in a stream mesocosm experiment. Freshwater Biology 2019, 64, 750-760.

Boulton, A.J.; Lake, P.S. Effects of Drought on Stream Insects and its Ecological Consequences. In Aquatic Insects: Challenges to Populations: Proceedings of the Royal Entomological Society's 24th Symposium; Lancaster, J., Briers, R.A., Eds.; CABI: Wallingford, United Kingdom, 2008; pp. 81–102.

  1. 270. Please, indicate the software used to perform the statistical analysis

We used Canoco 5 software as it is explicitly written in line 255. It enables the testing of “predictor explanatory power“ using non-parametric permutation statistics. It reshuffles cases = samples (taxa) and calculates individually “dependence of samples” on the predictors for each trial = permutation. Then it compares the original taxonomic assemblage with the set of permutations using so-called pseudo-F-statistic for an estimation of the significance of the original assemblage (Šmilauer and Lepš, 2014) 

Line 389-390. These sentences should be moved to the introduction section. Also, some sentences about macroinvertebrates as indicators of chemical pollution and how FFG can influence contaminants accumulation should be added in the introduction (i.e., https://doi.org/10.1016/j.scitotenv.2019.134282; https://doi.org/10.3390/biology9090288). 

Our response: We are not sure which line did you mean because all cited lines are shifted, and it does not match the version of this article, neither before nor after the initial adjustments after the editor’s comments. 

We added representants of susceptible aquatic organisms and higher taxonomic groups (lines 70–71) and we cited the particular literature which we properly read. From the articles which you suggested, we chose only one of them because the second one (doi:10.1016/j.scitotenv.2019.134282) was not matched well to our intro section – First, it relates to trace elements, second, the high correlations of Baetis sp. and Caenis sp. mayfly densities with La, Ce, Gd (as elements indicating anthropogenic activities) as the main result was not usable without changing the aim of our study (drought and insecticide disturbance) according to our opinion.

On the other hand, we can confirm the heptageniid sensitiveness to heavy metals reported in the article doi.org/10.3390/biology9090288. We observe the same response on data in preparation from the Litavka brook heavily contaminated by zinc, lead and other toxic metals. Therefore, we represented the heptageniid family as a susceptible group to anthropogenic disturbance (line 71).

  1. 618. Conclusion section should include few sentences. Please, reduce the section accordingly (only the key points should be reported). 

Our response: We removed this sentence (line 551–553):

“The shift in the community composition was significant along the gradient of strength of both types of disturbances in several ways, despite the low number of samples”.

We did not find any other redundant information.

New modified improvements

Line 14: We changed “response” to plural “responses”.

Line 70–71: We added examples of sensitive indicator species and higher taxonomic groups as a reaction to the recommendation of reviewer #2 with a particular citation (for more see our response above).

Line 98: We removed “and”.

Line 241: …“the organic pollution“ – “the“ was removed.

Line 250: “Fast flowing” was replaced for “Fast-flowing”.

Line 351 (Table 3): The title was enhanced with explanations: “C (unaffected, transect 1), 1 (affected, transect 1) and 2 (affected, transect 2)” but font size had to be decreased from 9 to 8 because of limited space.

Line 360: We added a comma before “in general”.

Line 362: We replaced “In addition” for “Besides”.

Line 528: We replaced “despite the fact that” for “although”.

Line 592–832: We adjusted both, the part “References” as well as the order of cited literature in square brackets inside the main text (line 39–543)

Round 2

Reviewer 1 Report

Thank you for responding to the comments. 

It is great that Dr. Michael Lockwood has agreed to edit your manuscript. As I mentioned in my first review, the introduction seemed like written by someone else or it did not go through Dr. Lockwood's editing. You need to look at it in more detail. I tried to straighten up some of the confusing terms and language used but the paper needs another close editing before publication. 

In my opinion, the sentence in Line 60 will benefit from editing. The way it is written now, you are establishing causality that does not exist. 

In Line 72 you use the term "natural experiment" because one author defined it before. By the explanation you've given, I agree that my suggestion of a "field experiment" or "large-scale" experiment is incorrect. However, the term "natural experiment" is not widely used. I suggest you use "natural attenuation" instead.

Some sections in the methods have been taken from Rocha et al. 2015. Monitoring of Pesticide Residues in Surface and Subsurface Waters, Sediments, and Fish in Center-Pivot Irrigation Areas. Paraphrase the methods, include an in-text citation, and add Rocha et al. paper to the reference section. Otherwise, the current text is considered plagiarism. 

Author Response

Dear reviewer,

we appreciate your suggestions and comments and we tried to solve everything carefully. Thank you for your time and effort.

Please, see our response below.

Yours faithfully,

Marek Let (on behalf of collective authors)

It is great that Dr. Michael Lockwood has agreed to edit your manuscript. As I mentioned in my first review, the introduction seemed like written by someone else or it did not go through Dr. Lockwood's editing. You need to look at it in more detail. I tried to straighten up some of the confusing terms and language used but the paper needs another close editing before publication. 

We sent it back to Dr. Lockwood and he made minor editing in the intro section (lines 40–86). In addition, we attach a confirmation letter, which Dr. Lockwood wrote for us, relating to the editing of the language use in our article.

In my opinion, the sentence in Line 60 will benefit from editing. The way it is written now, you are establishing causality that does not exist. 

We checked it again and we do not see any explanation of causality – only correlation expressed by the verb “to accompany”. It should express the occurence of common accompanying factors (co-factors – “followers”). We briefly named these factors without any expression or establishing of complicated causation relationships among them because their network can be very complicated, non-stable, site-specific in many points of view, although interdependent as well (e.g., see Blann et al., 2009).

We are convinced that you can accept that all these human-induced impacts, which are named in the particular sentence, are caused by anthropogenic activities. Pesticides are not responsible for channelisation or drainage; nonetheless, these factors often appear together, e.g. as a reaction of local management to adverse natural conditions and/or for enhancement production potential of a landscape to be more beneficial.

However, we used the word “often” instead of the “usually” (line 62) to reduce the certainty of our statement because we realize that our knowledge is of course very limited. We relied on the conditions in Central Europe that are familiar to us. But we also read a piece of evidence in the literature that these phenomena are also typical, e.g. in human-impacted Chinese big rivers – fluctuating water regime (Chen et al., 2001) accompanied by various types of contamination etc. Of course, damming can reduce an extend of fluctuations.

Furthermore, the particular sentence is just an expression of our experience and opinion; nevertheless, we support this expression with carefully cited literature sources that mention an identical point of view, explained in more detail. To have opposite opinions or doubts about this statement is ok, but for this case, we recommended reading particular cited sources.

To make it easier for you, we report a part of the article written by Blann et al. (2009) cited within our article as the number [44]:

“Effects of drainage on aquatic ecosystems include both direct and indirect effects. Direct effects include habitat loss due to stream channelization and conversion of wetlands to croplands. Indirect effects include water quality and habitat impacts of sediment, phosphorus, nitrogen and other contaminants in agricultural runoff, as well as hydrologic alteration in the form of altered volume and timing of runoff. Alteration of flow regimes in turn drives a complex of interrelated changes in stream morphology, instream and riparian habitats, nutrient cycles, and biota.”

Figure taken from Blann et al. (2009) – if you do not see it, please open the pdf. version of this response of us uploaded to the journal website.

There are much more articles that could have been cited – we used those which are cited in the given section (lines 60–64) [44-48].

Blann, K. L., Anderson, J. L., Sands, G. R., & Vondracek, B. (2009). Effects of agricultural drainage on aquatic ecosystems: a review. Critical reviews in environmental science and technology, 39(11), 909-1001.

Chen, X., Zong, Y., Zhang, E., Xu, J., & Li, S. (2001). Human impacts on the Changjiang (Yangtze) River basin, China, with special reference to the impacts on the dry season water discharges into the sea. Geomorphology, 41(2-3), 111-123.

In Line 72 you use the term "natural experiment" because one author defined it before. By the explanation you've given, I agree that my suggestion of a "field experiment" or "large-scale" experiment is incorrect. However, the term "natural experiment" is not widely used. I suggest you use "natural attenuation" instead.

We really wanted to comply with your suggestion. However, we briefly checked the internet and we did not find a more reasonable link between the macroinvertebrate ecology and the term “natural attenuation” than between the term “natural experiment” and the macroinvertebrate ecology.

            We apologize but this suggestion does not seem suitable. If you want to fix it, please, give us support information for it because we do not know the meaning of this word in the particular context. Therefore, we cannot simply accept this term.

“Natural attenuation”

https://scholar.google.cz/scholar?hl=cs&as_sdt=0%2C5&q=natural+attenuation+study+macroinvertebrate&btnG=

“Natural experiment”:

https://scholar.google.cz/scholar?hl=cs&as_sdt=0%2C5&q=natural+experiment+macroinvertebrate&btnG=

Some sections in the methods have been taken from Rocha et al. 2015. Monitoring of Pesticide Residues in Surface and Subsurface Waters, Sediments, and Fish in Center-Pivot Irrigation Areas. Paraphrase the methods, include an in-text citation, and add Rocha et al. paper to the reference section. Otherwise, the current text is considered plagiarism. 

We honestly apologise for this. It was caused by miscommunication between one external co-author who should be responsible for the correctness of this section and the rest of authors. We made our best to fix the problem (lines 150–200) and we added Rocha et al. (2015) [61] and another source, Ferenčík and Schovánková (2013) [60], into the particular part of the materials and methods section and into the reference section as well (lines 778–784).
